# Preparation and Characterization of Salsalate-Loaded Chitosan Nanoparticles: In Vitro Release and Antibacterial and Antibiofilm Activity

**DOI:** 10.3390/md20120733

**Published:** 2022-11-24

**Authors:** Sivarasan Ganesan, Jagadeesh Kumar Alagarasan, Mohandoss Sonaimuthu, Kanakaraj Aruchamy, Fatemah Homoud Alkallas, Amira Ben Gouider Trabelsi, Fedor Vasilievich Kusmartsev, Veerababu Polisetti, Moonyong Lee, Huang-Mu Lo

**Affiliations:** 1Department of Environmental Engineering and Management, Chaoyang University of Technology, Taichung 41349, Taiwan; 2School of Chemical Engineering, Yeungnam University, Gyeongsan 38541, Republic of Korea; 3Department of Physics, College of Science, Princess Nourah bint Abdulrahman University, P.O. Box 84428, Riyadh 11671, Saudi Arabia; 4Department of Physics, Khalifa University of Science and Technology, Abu Dhabi 127788, United Arab Emirates; 5Wallenberg Wood Science Center, Department of Fibre and Polymer Technology, School of Engineering Sciences in Chemistry, Biotechnology and Health, KTH Royal Institute of Technology, SE-100 44 Stockholm, Sweden

**Keywords:** chitosan nanoparticle, salsalate, drug release, antibacterial, cytotoxicity, antibiofilm

## Abstract

The controlled-release characteristic of drug delivery systems is utilized to increase the residence time of therapeutic agents in the human body. This study aimed to formulate and characterize salsalate (SSL)-loaded chitosan nanoparticles (CSNPs) prepared using the ionic gelation method and to assess their in vitro release and antibacterial and antibiofilm activities. The optimized CSNPs and CSNP–SSL formulation were characterized for particle size (156.4 ± 12.7 nm and 132.8 ± 17.4 nm), polydispersity index (0.489 ± 0.011 and 0.236 ± 132 0.021), zeta potential (68 ± 16 mV and 37 ± 11 mV), and entrapment efficiency (68.9 ± 2.14%). Physicochemical features of these nanoparticles were characterized using UV–visible and Fourier transform infrared spectroscopy and X-ray diffraction pattern. Scanning electron microscopy studies indicated that CSNPs and CSNP–SSL were spherical in shape with a smooth surface and their particle size ranged between 200 and 500 nm. In vitro release profiles of the optimized formulations showed an initial burst followed by slow and sustained drug release after 18 h (64.2 ± 3.2%) and 48 h (84.6 ± 4.23%), respectively. Additionally, the CSNPs and CSNP–SSL nanoparticles showed a sustained antibacterial action against *Staphylococcus aureus* (15.7 ± 0.1 and 19.1 ± 1.2 mm) and *Escherichia coli* (17.5 ± 0.8 and 21.6 ± 1.7 243 mm). Interestingly, CSNP–SSL showed better capability (89.4 ± 1.2% and 95.8 ± 0.7%) than did CSNPs in inhibiting antibiofilm production by *Enterobacter tabaci* (*E2*) and *Klebsiella quasipneumoniae* (*SC3*). Therefore, CSNPs are a promising dosage form for sustained drug delivery and enhanced antibacterial and antibiofilm activity of SSL; these results could be translated into increased patient compliance.

## 1. Introduction

In the current scenario of pharmaceutical research, polymers have been extensively used as active agents for drug delivery [1]. By establishing matrix, membrane, or nanocarriers, they can regulate drug release over an extended time and prevent repeated dosing. Because they can be crucial in the healing process, naturally functional and reactive polymers, such as chitosan (CS), guar gum, chitin, and alginate, are employed as drug delivery vehicles [2]. CS is a semi-crystalline and linear polysaccharide and an interesting functional biomaterial owing to its excellent biodegradability, low cost, biocompatibility, and low toxicity; therefore, it is an intriguingly active biopolymer [3]. Additionally, it has bio-adhesion qualities and antibacterial efficacy against a wide range of microorganisms, such as filamentous fungi, yeast, and bacteria [4]. The predetermined range of CS as a chelator to cellular trace elements, which results in toxicity and growth suppression, is another important aspect. CS displays its antibacterial effects in acidic pH only, which is influenced by several variables, such as the level of polymerization, molecular weight, chitin species, and solvent [5]. CS is a viable candidate for drug delivery in an encapsulated form to an infected area. This naturally occurring cationic polysaccharide possesses mucoadhesive properties that enable its transport across the mucosal membrane [6]. The usage of CS as a carrier for a slow and sustained release of drugs can address the issue of drug half-life. However, further studies on the controlled release efficacy of CSNPs as drug carriers are necessary [7,8].

Anti-inflammatory drugs fall into two primary types, corticosteroids and non-steroidal anti-inflammatory drugs (NSAIDs), which reduce enzymatic activity [9]. Since arachidonic acid is a recognized source of prostaglandins, NSAIDs work pharmacologically by suppressing the activities of certain enzymes, while corticosteroids pharmacologically prevent the release of arachidonic acid [10]. NSAIDs are most frequently recommended by doctors in contemporary medicine despite their undesirable side effects owing to their demonstrated usefulness in decreasing fever, reducing inflammation, and alleviating pain [11]. Salsalate (SSL), an NSAID, can be used to treat a variety of disorders involving pain [12]. However, it can cause some severe side effects, including high blood pressure, dizziness, upset stomach, and nausea. Therefore, using a low dosage of this drug may be fruitful in reducing these side effects [13,14]. A solution for this concern is reducing the particle size of this drug to micron or submicron, which can be easily released and adsorbed in the body even if a low dosage of this drug is used in the tablet [15]. Salicylate not only affects mammals but also causes physiological and morphological changes in bacteria [16,17]. Antimicrobial drug therapy is determined by several factors, such as bacteriostatic versus bactericidal mechanisms, the spectrum of activity, dosage, route of administration, and potential side effects. There are also potential drug interactions to consider. It depends on the interaction between the antibacterial drug and the target bacteria, whether it behaves like a bacteriostatic or a bactericidal agent [18]. When bacteria are treated with bacteriostatic drugs, they inhibit their growth reversibly, allowing bacteria to begin growing again once they are removed from the body [19]. Compared to bactericidal drugs, which kill their target bacteria, antibiotics do not cause any side effects. The antimicrobials that target narrow-spectrum bacteria target only specific subsets. A few narrow-spectrum drugs target only Gram-positive bacteria, while others focus exclusively on Gram-negative bacteria. To minimize collateral damage to the normal microbiome, it is best to use narrow-spectrum antimicrobials to treat an infection caused by an identified pathogen [20]. There are a variety of pathogens that are treated with broad-spectrum antibiotics, including both Gram-positive and Gram-negative pathogens. Broad-spectrum antibiotics are often used as empiric therapies while awaiting laboratory identification of the underlying pathogen [21]. For example, it affects protein expression in *Escherichia coli* and *Staphylococcus aureus*. Salicylate has also been linked to the decrease in tolerance to several antimicrobial drugs and the generation of substances that contribute to bacterial pathogenicity. Nano-encapsulation of medicinal drugs (nanomedicines) is beneficial in improving internal absorption, solubility, and storage time [22], and therefore, the cost of drug use and the risk of toxicity for the patient could be reduced.

CS is effective against bacteria, viruses, and fungi [23]. CS has bactericidal properties, which result from an electrostatic contact between its NH_3_^+^ groups and phosphoryl groups of phospholipids found in the cell membranes of bacteria [24]. As a result, pores form on the bacterial cell membrane to make them permeable, which ultimately ruptures bacterial cells. However, owing to the loss of positive ions on the amine groups while becoming insoluble at neutral and basic pH, the antibacterial action of CS is restricted at acidic pH only. Despite antibiotic therapy, bacterial infectious illness continues to cause a sizable amount of mortality worldwide. The lack of efficient biocompatible delivery systems for the majority of hydrophobic treatments and antimicrobial resistance is a significant issue for bacterial infection therapy [25]. The concentration, level of deacetylation, molecular weight, pH, and types of microorganisms exposed to the biopolymer are the key determinants of its antibacterial action. 

Polymeric nanoparticles (NPs) and microparticles can transport drugs to the lungs under regulated conditions with sustained drug release to extend the duration of action, lessen adverse effects, and enhance patient compliance [26]. Owing to their increased surface area and a strong propensity for a quantum-size effect within bacterial cells, chitosan nanoparticles (CSNPs) have the potential to rupture bacterial cell membranes, allowing internal components to seep out and ultimately causing bacterial death [27,28,29]. CSNPs have excellent drug encapsulation efficiency (EE) and a variety of release characteristics, making them suitable for delivering various drugs in different settings [30]. CSNPs are prepared using several techniques, including coacervation, emulsion-droplet agglomerates, inverse micelles, ionic gelation, and molecular self-assembly alteration [8]. Ionic gelation techniques have been extensively studied to prepare drug-loaded CSNPs [31]. The ion gelation process can be used to prepare CSNPs using tripolyphosphate (TPP) as a crosslinker [32]. In a typical non-covalent interaction, ionic crosslinking of CS can be achieved by associating with the multivalent negatively charged ions of TPP [33]. In contrast, the ionic gelation method offers the advantage of simplicity with no requirement for complicated equipment. They use electrostatic crosslinking rather than chemical crosslinking, which lessens the possibility of the particles causing hazardous consequences.

In the present study, we synthesized CSNP–SSL, using an ionic crosslinking method, as a potential carrier of SSL to improve in vitro release and antibacterial and cytotoxicity activity. The physicochemical properties of CSNP–SSL were investigated using Fourier transform infrared (FTIR), X-ray diffraction (XRD), scanning electron microscopy (SEM), transmission electron microscope (TEM), and in vitro drug release, and drug loading capability was monitored. The antimicrobial activity of CSNPs and CSNP–SSL against Gram-positive and Gram-negative bacteria, such as *S. aureus* and *E. coli,* was assessed. Furthermore, the antibiofilm activity of CSNPs and CSNP–SSL was studied against *Enterobacter tabaci* (*E2*) and *Klebsiella quasipneumoniae* (*SC3*).

## 2. Results and Discussion

### 2.1. Encapsulation Efficiency

The quantity of SSL was kept unchanged while CS concentrations were varied to assess the EE% of CSNP–SSL. The EE increased from 56.7 ± 1.21% to 68.9 ± 2.14% when the amount of CS was increased from 0.2% to 1.0% (*w*/*w*) [34] (Table 1). Owing to its low water solubility, high entrapment (68.9 ± 2.14%) of SSL within CSNPs was effective for drug purposes. The corresponding amino groups increased with increasing CS concentration, resulting in greater crosslinking with SSL and more encapsulation [35]. At a constant TPP concentration, encapsulation improved when CS concentration increased to 1.0%. Therefore, this composition with a CS concentration of 1.0% (*w*/*w*) was used for subsequent experiments. The above findings were in good agreement with the previous work to encapsulate drug molecules into the CSNPs [30,34,36,37,38,39].

### 2.2. Mean Particle Size, Polydispersity Index (PDI), and Zeta Potential

In the present work, CSNPs were synthesized using the ionic gelation process; both before and after SSL drug formulations were loaded to assess the antibacterial and antibiofilm activities of CSNPs and CSNP–SSL [40]. For CSNPs and CSNP–SSL, hydrodynamic diameters were 156.4 ± 12.7 nm and 132.8 ± 17.4 nm; PDI values were 0.489 ± 0.011 and 0.236 ± 0.021; and zeta potentials were 68 ± 16 mV and 37 ± 11 mV, respectively. The mean zeta potential of the formulated CS NP-SSL decreased with the addition of the SSL drug. The size of CSNPs and the PDI have significant changes when combined with SSL. Since the electrostatic repulsion between the NPs is what causes the zeta potential of CSNPs and CSNP–SSL to be higher than the value of 30 mV, it was also known that these NPs had good stability [33]. This outcome was consistent with earlier findings that CSNPs have grown in particle size, PDI, and zeta potential. Additionally, unlike large particles, small particles tend to accumulate at the tumor site according to the improved permeability and retention impact [41].

### 2.3. Optical Properties of SSL-Loaded CSNPs

Optical properties of CSNPs, SSL, and CSNP–SSL were studied using UV–vis spectrophotometry. The UV–vis spectrum of CSNPs showed a broad absorption band at 334 nm, which corresponds to the n–π* transition of CSNPs [42] (Figure 1). Moreover, the absorption bands of SSL at 305, 240, and 215 nm corresponding to the π–π* transition [43]. Compared with that of CSNP–SSL, the characteristic peak of SSL showed a blue shift from 305 to 276 nm and a small red-shifted hump at 344 nm, marked as a round shape in Figure 1. This blue shift was owing to the decrease in the size of CSNPs with maximum absorbance wavelength [44]. The results indicate that SSL was successfully encapsulated within CSNPs.

### 2.4. Surface Properties of SSL-Loaded CSNPs

Figure 2 depicts the FTIR spectra of CS, CSNPs, SSL, and CSNP–SSL. The results demonstrated the basic structural features of CS at 3431, 2919, and 2869 cm^−1^ assigned to –OH, –NH_2_, and –CH stretching, respectively. Moreover, –NH_2_, C–O–C, and pyranoside ring stretching vibration corresponded at 1599, 1082, and 599 cm^−1^, respectively [37] (Figure 2a). In contrast, characteristic peaks of CS appeared/disappeared or shifted to 3315, 2884, 1584, 1421, 1067, and 712 cm^−1^ when compared with those of corresponding CSNPs [45] (Figure 2b). These results affirm that CSNPs without any encapsulation can be of significant benefit. These peaks in the FTIR spectra of CSNPs shifted hypsochromically to 1641 and 1562 cm^−1^ because of the interaction between the NH_3_^+^ groups of CS and phosphate groups of TPP [38]. In a prior investigation, similar outcomes of the development of CSNP-treated TPP were described [34]. FTIR spectrum of SSL is shown in Figure 2c. The characteristic peaks of SSL appeared at 3463, 3205, 2815, 2656, 1733, 1673, 1614, 1485, 1409, 1259, and 745 cm^−1^. The structural characteristics of SSL-loaded CSNPs were disclosed using FTIR findings [43] (Figure 2d). It further demonstrates the comparison between SSL unloaded and loaded CSNPs. The bonds in CS, including hydrogen bonds, van der Waals forces, and dipole moments, are often represented by little variations in absorption peaks and cause the sugar ring to expand. Owing to the presence of SSL, the measured peaks between 3600 and 2800 cm^−1^ altered. A further indication of substantial physical interactions between SSL and CSNPs was the observation of peaks at 1655, 1431, 1279, 1135, 947, 829, 676, and 539 cm^−1^ of SSL-loaded CSNPs [46].

Figure 3 displays the XRD patterns of CS, CSNPs, SSL, and CSNP–SSL for the 2θ range of 10–80°. In contrast to CSNPs that have both peaks at 2θ of 11.8° and 20.7°, CS exhibited two peaks at 2θ of 17.1° and 24.6° (Figure 3a,b), indicating a significant degree of amorphous phase [47]. The basic diffraction peaks were displaced, and peak strength decreased after crosslinking of CS with TPP (Figure 3b). The XRD pattern of sharp diffraction peaks of SSL appeared at 13.1°, 15.8°, 18.7°, 21.3°, 22.8°, 23.7°, 27.4°, and 49.4° 2θ, as shown in Figure 3c [43]. Furthermore, new peaks appeared at 2θ of 19.3°, 23.4°, and 40.9°, indicating that the amorphous phase was induced by ionic interaction between SSL and CSNPs [48] (Figure 3d). According to the XRD patterns, CSNP–SSL either took the form of a molecular or amorphous dispersion within the CS matrix. In conclusion, the removal of distinctive crystalline peaks of SSL was consistent with earlier studies because SSL was completely coated and encased by CSNPs, or it may be changed into an amorphous condition.

### 2.5. Morphology of SSL-Loaded CSNPs

SEM images were captured for CSNPs without and with SSL (Figure 4). The images showed that CSNPs had a spherical morphology, and most of the particles were aggregated with a diameter between 500 nm and 1 μm, as shown in Figure 4a–c [49]. As shown in Figure 4d–f, the SEM image of SSL-loaded CSNPs showed uniform and spherical morphology with a diameter between 200 and 700 nm, showing an effect of aggregation between CSNPs and SSL [50]. It suggested possible stabilization of CSNPs owing to positive surface charges. TEM images of CSNPs without and with SSL are illustrated in Figure 5. The morphology of CSNPs was spherical, and the surface was smooth (Figure 5a). As seen in Figure 5b, the particle size of CSNP–SSL decreased following the addition of SSL. As shown in Figure 5c,d, CSNPs and CSNP–SSL composites with average particle sizes of 119.3 ± 3.2 nm and 71.4 ± 1.27 nm, respectively, were obtained, indicating that the size decreased the diameter of SSL-loaded CSNPs [51,52]. CSNP–SSL composites were positively charged, as was evident by zeta potential values.

### 2.6. Drug Release and Kinetics

Sudden and sustained release are the two possible drug release profiles of NPs. Drugs adsorbed onto the surface of NPs produce the initial burst phase. A gradual diffusion of the drug from the polymer matrix and the breakdown of the carrier matrix is responsible for sustained drug release [53]. The in vitro release and kinetics study of SSL from CSNPs using phosphate buffer of pH 7.4 for 48 h is depicted in Figure 6. The released percentage was calculated concerning the encapsulated SSL concentration. In the first phase, there was an initial burst release of around 64.2 ± 3.2% at 6 h [54], followed by a slow release from 6 to 24 h when around 84.6 ± 4.23% of the SSL drug was released [2], as shown in Figure 6a. The burst release of the drug could be due to the dissolution of the surface SSL drug that is poorly entrapped in the polymer matrix, and slow release is due to diffusion of the SSL drug present in the core of CSNPs. The drug release was observed to be prolonged by CSNPs in comparison to that by the SSL solution. As seen in Figure 6b, the SSL concentrations increase dramatically over 6 h. A recent study on drug release kinetics [55] suggests that the release response follows first-order kinetics. In the first 6 h of our experiment, when the release of SSL from CSNP-SSL is the main process, we also investigated the release kinetics. With an R^2^ value of 0.9761 in the CSNP-SSL, the rate of dissolution and concentration of the SSL medication obeyed first-order kinetics. Based on the results, it may be conceivable to develop CSNP-SSL into a potent tool for biomedical applications that need the controlled release of medications into a biological system over an extended time.

### 2.7. Antibacterial Activity

CS is widely used as an antibacterial agent because of its broad antimicrobial action, non-toxicity, and excellent biodegradability [56]. Antibacterial efficacies of drugs and NPs have been described in numerous studies. Here we used the agar diffusion method to explore the antibacterial properties of CSNPs, SSL, and CSNP–SSL against *S. aureus* and *E. coli* (Figure 7). The test CSNPs, SSL, and CSNPs-SSL were all necessary to determine the antibacterial activity as measured by the diameter of the growth inhibition zone. The ratio of disk area to the contact area, inoculum, and type of solid media are the key determinants of the shape and size of the clear zone in the disk method.

When tested against *S. aureus* and *E. coli*, CSNPs and CSNP–SSL showed inhibitory zones with mean diameters of 15.7 ± 0.1 and 19.1 ± 1.2 mm, and 17.5 ± 0.8 and 21.6 ± 1.7 mm, respectively. *S. aureus* and *E. coli* had zones of inhibition by SSL (80 mg) with diameters of 11.2 ± 0.1 and 13.7 ± 0.9 mm, respectively. Higher antibacterial activity was shown by CSNPs and CSNP–SSL than by SSL drug [57,58] (Figure 7). CS itself possesses antibacterial activity because of its cationic properties [59]. Positively charged CS attaches to the negatively charged bacterial cell membrane, thereby increasing the antibacterial activity of CSNP–SSL. To the best of our knowledge, the antibacterial activity of CSNP–SSL has not yet been explored.

### 2.8. Antibiofilm Activity

SEM was used to examine how morphogenesis affects the general structure of anti-biofilm. *E. tabaci* (*E2*) and *K. quasipneumoniae* (*SC3*), two mixed-species antibiofilm, were exposed to CSNPs and CSNP–SSL for 72 h [60] with the mixed-species biofilms grown on filter sheets; SEM was utilized to evaluate the anti-biofilm properties of CSNPs and CSNPs–SSL [61]. Dense antibiofilm development was seen in the untreated samples, exhibiting typical antibiofilm growth on the abiotic surfaces [62] (Figure 8). Gram-negative bacteria produced more antibiofilm than Gram-positive bacteria.

The majority of cell membranes displayed hazy borders and collapsed morphology after receiving control treatment for 1 h. In contrast, 100 μg/mL CSNP–SSL significantly reduced antibiofilm development. Antibiofilm resistance depends on the mode of antibiofilm growth and growth rate. When exposed to CSNP–SSL, certain cells were disrupted and shrank because of intercellular damage [63]. However, CSNPs at the same concentration had no discernible impact on the shape of cells (Figure 8). With CSNP–SSL at μg/mL, eradication rates were 89.4 ± 1.2% and 95.8 ± 0.7% for antibiofilm-grown *E. tabaci* (*E2*) and *K. quasipneumoniae* (*SC3*) colonizing the implant samples, respectively. This antibacterial activity was highly significant (*p* = 0.05) (Figure 8). Finding out whether a combination of extracellular components from bacteria and fungi results in a more viscous matrix and whether this explains the enhanced antibacterial resistance of mixed-species antibiofilm will be interesting [64]. Mixed-species reduction of antibiofilm growth is, therefore, a crucial discovery for alternative antibacterial drugs.

## 3. Materials and Methods

### 3.1. Materials

Salsalate (MW; 258.23, purity ≥ 98%) was purchased from Sigma (St. Louis, MO, USA) and dissolved in dimethyl sulfoxide (DMSO). Chitosan (50–190 kDa, Degree of deacetylation 75–85%) and sodium tripolyphosphate (MW; 367.8, purity, ≥98%) were obtained from TCI chemical reagent Co., Ltd. (Seoul, ).

### 3.2. Preparation of SSL-Loaded CSNPs

The ionic gelation process with a few minor modifications [36,65] was used to synthesize CSNPs. In brief, 0.5% (*w*/*v*) CS was dissolved in a 1% (*v*/*v*) acetic acid solution, and its pH was adjusted to 6.5 using NaOH. SSL powder was precisely measured and mixed with deionized water (DW) to prepare a solution of 50 mg/mL concentration. The SSL solution was added dropwise to the chitosan solution (CS and SSL ratio of 1:1) and was thoroughly mixed for 30 min with a probe sonicator (Q500 sonicator; 750 W, 20 kHz, diameter ~ 7 mm). The combination was then added to 2.5 mL of 1.5 mg/mL of (2% *w*/*v*) sodium tripolyphosphate (TPP) solution by stirring (JSHS-18A Analog hotplate Stirrer) at 600 rpm at 37 °C. The CSNP–SSL was spontaneously generated under probe sonication after the addition of TPP. After centrifuging (Avanti J-26S Series) the resulting NPs at 10,000 rpm for 15 min at 4 °C, the supernatant was discarded, and the CSNP–SSL was subsequently dissolved in DW. The same process was used to prepare CSNPs without SSL (Figure 1).

### 3.3. Characterization

A zetasizer (Particle Sizing Systems, Port Richey, FL, USA) was used to measure the particle size, PDI, and zeta potential of CSNPs and CSNP–SSL at 37 °C. Each measurement was repeated thrice, and the results are presented as mean ± standard deviation (SD). Each sample was properly diluted with DW before testing. UV–vis spectrophotometer was used to assess optical characteristics (Model Perkin Elmer Lambda). The FTIR spectra were collected over a range of 400–4000 cm^−1^ using Perkin-Elmer (Spectrum-RX1, USA) FTIR spectroscopy. An X-ray diffractometer (Shimadzu Corporation, Kyoto, Japan) was used to conduct the XRD studies in 10–80° range using CuKα as the radiation source (k = 1.5418) produced at 30 kV and 30 mA. SEM was used to observe the morphologies of CSNPs and CSNP–SSL (SEM, Jeol FESEM JSM-7600F). TEM Images from a Tecnai F-12 JEOL-JEM 2100 TEM were captured at a 200 kV accelerating voltage. CSNPs and CSNP–SSL samples were homogeneously mixed with DW in an ultrasonic bath before being poured onto air-dried copper grids placed on filter paper and then dried at room temperature. The particle diameter size distribution versus frequency percentage of CSNPs and CSNP-SSL was calculated by measuring 50 particles with 100 m^2^/g specific surface area using ImageJ software (National Institutes of Health, Bethesda, MD, USA).

### 3.4. Encapsulation Efficiency

The EE of CSNP–SSL was assessed using the ultrafiltration centrifugation technique [36]. CSNP–SSL was centrifuged for 15 min at 4 °C at 10,000 rpm. The amount of free SSL was calculated by measuring the absorbance of the supernatant at 305 nm using a UV–vis spectrophotometer. The EE% was obtained by deducting free SSL from the total amount of SSL added using the formula below:EE% = Total amount of SSL − the amount of free SSL/Total amount of SSL × 100(1)

### 3.5. In Vitro Drug Release Study

CSNPs were resuspended in an SSL-free solution, and the concentration of SSL was measured at specified intervals to determine the rate of SSL delivery by CSNPs. Dispersion of SSL in CSNPs was measured using UV–vis spectroscopy at 305 nm. For this, 25 mL of phosphate-buffered saline (PBS, pH 7.4) containing 20 mg of CSNPs was added, and the mixture was shaken at 60 rpm at 37 °C. At predefined intervals, 1 mL of CSNP solution was collected, and an equal amount of fresh PBS was added to the mixture. The sample was centrifuged, and the quantity of SSL in the supernatant was determined using spectrophotometry at each subsequent time point (0, 0.5, 1.0, 1.5, 2.0, 2.5, 3.0, 3.5,5.0, 6.0, 9.0, 12, 18, 24, 30, 36, 42, and 48 h). The proportion of SSL was determined using a standard curve. Further, to understand the dissolution behavior of SSL drugs, the percentage at which the release becomes controlled can be determined for each SSL drug. SSL dissolution kinetics in PBS 7.4 buffer were investigated. Each experiment was performed thrice.

### 3.6. Evaluation of Antimicrobial Activity

The agar well diffusion method was used to assess the antibacterial activity of CSNP–SSL. In the agar diffusion method, wells were created in the Mueller Hinton Agar (MHA) plate inoculated with the test bacteria using a cork borer. The wells were then filled with 50 μL of each of the three solutions: SSL-CSNP formulation, SSL (pH = 7), and ampicillin as the positive control. The agar medium was sterilized for 30 min at a pressure of 6.8 kgf/m^2^ (15 lbs). In an air-flow laminar chamber, this material was transported into sterilized Petri dishes. Cultures of *S. aureus* and *E. coli* (50 μL) were applied to the solid media. The diameters of the inhibitory zones were measured in millimeters (mm). Each experiment was carried out in triplicate, and all plates were incubated for 24 h at 37 °C.

### 3.7. Evaluation of Antibiofilm Activity

*E. tabaci E2* and *K. quasipneumoniae SC3* blended antibiofilm cells were cultured for 24 h at 37 °C with CSNPs and CSNP–SSL solution. To compare 3D cell culture with CSNPs and CSNP–SSL, cells (10^6^ CFU/mL) cultured on tissue culture polystyrene interfaces (96-well plates) were washed twice in PBS and incubated for 1 h with SYTO 9 stain. The bacterial cells were prefixed with glutaraldehyde and formaldehyde (2.5% and 2%, respectively) for 24 h at 4 °C, followed by post-fixation with 2% osmium tetroxide. After incubation, the plates were dried at 55 °C for 1 h before labeling, the used solution and cells were carefully disposed of, and weakly adhering cells were eliminated by washing twice with sterile 0.1 M PBS. The morphology of antibiofilm activity on the control, CSNPs, and CSNP–SSL was observed using SEM (JEOL JSM-5600LV). The assay was performed in triplicate and repeated at least thrice.

### 3.8. Statistical Analysis

Data were analyzed using OriginPro 8.5. The results were obtained from each experiment performed in triplicate, and they were presented as mean ± SD. A *p*-value < 0.05 was considered statistically significant.

## 4. Conclusions

CSNPs are widely used in biomedical applications, particularly in drug delivery and antibacterial systems. In the current study, the ionic gelation method was used to prepare CSNPs before and after conjugation with SSL by crosslinking using TPP to investigate the effect of antimicrobial and antibiofilm activities of CSNPs and CSNP–SSL. CSNPs and CSNP–SSL were characterized using PDI, zeta potential, FTIR, XRD, SEM, and TEM techniques. The morphology of CSNPs and CSNP–SSL were observed using TEM, and the CSNPs and CSNP–SSL were spherical with average particle sizes of 119.3 ± 3.2 nm and 71.4 ± 1.27 nm, respectively. SSL-loaded CSNPs were spherical with encapsulation efficiencies of approximately 68.9 ± 2.14%. In vitro release showed a burst release of 64.2 ± 3.2% SSL within the first 18 h, followed by a sustained release reaching 84.6 ± 4.23% after 48 h. Moreover, CSNPs and CSNP–SSL showed excellent antibacterial activity against *S. aureus* (15.7 ± 0.1 and 19.1 ± 1.2 mm) and *E. coli* (17.5 ± 0.8 and 21.6 ± 1.7 mm) by causing the release of intracellular components destroying the integrity of bacterial cell membranes. Finally, CSNP–SSL was capable of suppressing mixed-species antibiofilm of *E. tabaci* (*E2*) and *K. quasipneumoniae* (*SC3*) at 89.4 ± 1.2% and 95.8 ± 0.7%, which is a significant finding for alternative antimicrobial medicines. Therefore, SSL-loaded CSNPs have great potential for further clinical use as a delivery carrier and antimicrobial agent.

## Data Availability

Not applicable.

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
