# Peer review of "Preparation and Characterization of Salsalate-Loaded Chitosan Nanoparticles: In Vitro Release and Antibacterial and Antibiofilm Activity"

_marinedrugs, 2022, doi:10.3390/md20120733_

Round 1

Reviewer 1 Report

In this work the authors report the Preparation and Characterization of Salsalate-Loaded Chitosan Nanoparticles: In vitro Release and Antibacterial and Biofilm Activity. Overall the work is meaningful, the review deserves to be published, however some points should be further clarified, and therefore minor revisions may be considered:

Comments

1.      The optimized CSNPs and CSNP–SSL formulation were characterized for particle size, polydispersity index, zeta potential, and entrapment efficiency. The author should provide the above study values.

2.      The author should provide some quantitative information in the abstract section.

3.      The author should cite suitable references in the following sentences. CS is a viable candidate for drug delivery in an encapsulated form to an infected area. This ionic polymer can travel across the mucus layer owing to its muco-adhesive characteristics.

4.      Replace this word “medications” in the entire manuscript.

5.      The authors are encouraged to provide all the materials used to perform the experiments and analysis, along with their respective suppliers. Moreover, Please report the molecular weight and deacetylation degree of chitosan; this is relevant information to ensure appropriate reproducibility.

6.      The author should revise the following sentence. With the introduction of SSL, the mean zeta potential of CSNP–SSL decreased.

7.      Explain it. What is the meaning of the following sentence? It further exemplified how SSL-loaded and unloaded CSNPs differed from one another.

8.      The author reported “aggregated with a diameter between 500 and 1000 nm as shown in Figure 4 (a–c)”. 1000 nm should be 1 μm.

9.      As for Conclusion, the author may further clarify applications perspectives of the current work.

Reviewer 3 Report

This is a study on the preparation of salsalate-loaded chitosan nanoparticles, their characterization by various techniques, study of controlled release of the drug contained and their antibacterial/biofilm control testing. There are several modifications, clarifications and further details needed before the manuscript can be accepted for publication, ranging from very minor things to more important issues.

Abstract punctuation: there is an inappropriate and excessive use of semicolons (";") in places where simple commas (",") would have been better suited

Line 53: citation needed for "muco-adhesive characteristics"

Lines 65-66: citations needed for the side-effects enumerated here

Line 102-103: singular verbs needed here: "does not require", "it uses"

The images (SEM, TEM) in Scheme 1 already show some results obtained, therefore this scheme and its discussion belong to the Results and Discussion section of the manuscript, not the Introduction

Line 121: There is no Table 1 in the pdf version I downloaded from the platform to perform this review. Where is Table 1 and what does it contain?

Lines 124-125: To me, if 56.7% of SSL is encapsulated with 0.2% CS provided, then an increase to 68.9% encapsulation with an increase to 1% CS added is not truly an increase in encapsulation efficiency per nanoparticle; there are simply more nanoparticles for the same amount of SSL, therefore there will obviously be less SSL left non-encapsulated; given that the amount of CS increased five times and the percent encapsulation only by 12%, I would argue that the efficiency of encapsulation is actually worse per nanoparticle than before 

Line 135: were (not "was") observed

Line 162: that is "hypsochromically" (not "hypo-"); the wavelength changes, not just the intensity of absorption

Line 165: please cite the prior investigation you refer to here

Line 166: FTIR spectrum (singular)

Line 220: Figure 6 (not 2)

Line 222: What was the criterion for calling 18 h "the burst release phase"? From the figure there is a sudden increase ("burst") in only the first 6 hours, then it slows down significantly, until becoming steady beyond 18 hours

Figs. 4-5: How were the frequencies in Figs. 4c, 4f, 5c, and 5d measured? What was the sampling area and from how many replicates? These details are not provided in the Materials and methods section.

Materials and methods: Please indicate details (type, manufacturer) for the probe sonicator, stirrer, centrifuge, FTIR spectrometer

The international symbol for "mililiter" is "mL" (L capitalized) and should appear as such throughout the manuscript

Some acronyms are not defined anywhere in the manuscript: DMSO, MHA, SYTO, etc. Please define them at the time of first occurrence or provide a list of abbreviations.

Section 3.4 VS Section 3.5: Why for encapsulation efficiency assessment free SSL in supernatant is measured spectrophotometrically at 305 nm, but during the in vitro release study, the same drug is measured at 320 nm? Shouldn't the absorption band be the same for the free drug?

Line 330: perhaps "kgf m-2" for it to be a pressure unit (kilogram-force per square meter)

Round 2

Reviewer 3 Report

All previously raised issues were addressed. I have no further comments.